# Type 2C Protein Phosphatase MoPtc6 Plays Critical Roles in the Development and Virulence of *Magnaporthe oryzae*

**DOI:** 10.3390/jof11050335

**Published:** 2025-04-24

**Authors:** Frankline Otieno Jagero, Abah Felix, Yakubu Saddeeq Abubakar, Meilian Chen, Wilfred M. Anjago, Hatungimana Mediatrice, Nkurikiyimfura Oswald, Osakina Aron, Wei Tang, Zonghua Wang, Jules Biregeya

**Affiliations:** 1Fujian Universities Key Laboratory for Plant-Microbe Interaction, College of Life Science, Fujian Agriculture and Forestry University, Fuzhou 350002, China; frankja23@gmai.com (F.O.J.); fabah11@gmail.com (A.F.); ay.saddeeq@yahoo.com (Y.S.A.); osakina.aron@yahoo.com (O.A.); tangw@fafu.edu.cn (W.T.); 2Fuzhou Institute of Oceanography, Minjiang University, Fuzhou 350108, China; meilian2019@mju.edu.cn; 3College of Plant Protection, Fujian and Agriculture and Forestry University, Fuzhou 350002, China; mabechewilfred@gmail.com (W.M.A.); mediatunga@gmail.com (H.M.); nk.oswaldo@gmail.com (N.O.)

**Keywords:** *Magnaporthe oryzae*, phosphorylation, protein phosphatases, stress tolerance, virulence

## Abstract

Rice blast caused by *Magnaporthe oryzae* pathotype is the worst disease that leads to serious food insecurity globally. Understanding rice blast disease pathogenesis is therefore essential for the development of a blast disease mitigation strategy. Reverse phosphorylation mediated by phosphatases performs a vital function in the activation of diverse biological mechanisms within eukaryotic. However, little has been reported on the roles of PP2Cs in the virulence of blast fungus. In this current work, we deployed functional genomics and biochemical approaches to characterize type 2C protein phosphatase MoPtc6 in blast fungus. Deletion of *MoPTC6* led to a drastic reduction in conidiophore development, conidia production, hyphal growth, and stress tolerance. Western blotting assay demonstrated that the phosphorylation level of MoOsm1 was decreased while MoMps1 was increased in the MoPtc6 deletion mutant, and comparative transcriptome assay revealed a higher number of expressed genes between mutant and wild type. Localization assay confirmed that MoPtc6 is sub-localized in the cytoplasm of mycelia, spores, and in the appressoria of *M. oryzae*. Furthermore, disruption of *MoPTC6* impaired appressoria turgor pressure and glycogen utilization; more findings revealed attenuation of hyphal penetration and virulence upon deletion of *MoPTC6*. Generally, present findings suggested the role of MoPtc6 in the growth and virulence of *M. oryzae*.

## 1. Introduction

*Magnaporthe oryzae* is a filamentous phytopathogen that infects many cereals worldwide, especially rice [1]. Due to its adaptability and amenability to genetic manipulation, *M. oryzae* has been chosen by several researchers as a model fungal-plant pathosystem [2]. Conidia are formed from conidiogenous cells by a blastic mode of development, infection begins when a three-celled conidium lands and attaches to the hydrophobic surface of a rice leaf. The spore germinates, producing a long, narrow germ tube that differentiates into an appressorium, which subsequently penetrates and invades the plant cuticle [3]. Blast disease affects rice production, especially in temperate areas, and has already been detected in more than eighty countries, including China, India, Brazil, Bolivia, Indonesia, and Sri Lanka [4].

Type 2C protein phosphatases (PP2Cs) are potent serine/threonine protein phosphorylation and dephosphorylation factors and play important functions in the regulation of diverse cellular biological mechanisms [5]. PP2C proteins belong to the family of metal-dependent protein phosphatases (PPMs) that orientate towards several metal ions, including Mn^2+^, Mg^2+^, and Ca^2+^ [6,7,8]. These metal ions are situated in the middle of the enzyme to link up to the phosphorus atoms and improve the precision of the dephosphorylation assay [9].

In yeast, PP2C proteins are reported to perform indispensable roles in diverse biological processes and pathways, such as cell wall maintenance by mediating Slt2 functions, high osmolality glycerol, rapamycin functions, and cell wall remodeling pathways [10,11,12,13]. Specifically, *PTC6* is required in the activation and regulation of Slt2 during the dephosphorylation and vacuole formation under various stress factors, such as cell wall stress drugs in yeast cells [11]. Early findings in fungal pathogens, including *Fusarium graminaerum*, *Magnaporthe oryzae*, *Fusarium oxysporum*, and *Botrytis cinerea*, revealed that PP2Cs are important in fungal pathogenesis [14,15,16]. Previous reports disclosed that Ptc6 is involved in the pathogenicity and cell wall conservation in *F. oxysporum* [15]. Disruption of *PTC1* and *PTC2* affected vegetative growth and virulence in *B. cinerea* [16]. In *F. graminearum*, loss of Ptc1 decreased virulence in wheat at diverse growth stages [17]. In *M. oryzae*, *PTC5* and *PTC7* regulate MoMps1 and MoOsm1 phosphorylation levels, cell wall integrity, appressorium turgor generation, and glycogen mobilization [18]. Moreover, rice blast *PTC1* and *PTC2* deletion mutants influenced appressorium morphogenesis and growth, and ultimately reduced pathogenicity [19,20]. Recently, *PTC1* and *PTC2* were demonstrated to influence conidia formation, seed infection, and aflatoxin synthesis by regulating phosphoglycerate kinase1, and autophagic vesicle development in *Aspergillus flavus* [21]. Thus, these findings suggest that the functions of PP2Cs are highly related to the development and pathogenesis of diverse model fungi [22,23,24,25]. In this study, we examined the functional roles of MoPtc6 in relation to the growth and virulence of *M. oryzae*. The results demonstrated that Ptc6 mutants have weakened conidia production ability, perturbed vegetative growth, cell wall development, osmotic and oxidative stress tolerance, and regulation of MoMps1 and MoOsm1 phosphorylation. Taken together, findings provided evidence that the protein phosphatase MoPtc6 participates in fungal virulence and development.

## 2. Materials and Methods

### 2.1. Strain and Culture Conditions

Pathogenic fungus *M. oryzae* Guy11 was acquired from the fungal Genetics Stock Centre (FGSC 9462) and was employed to generate mutants in this work. The WT Guy11, ∆*Moptc6*, and ∆*Moptc6_C* were spotted on conidiation-inducing media, rice bran media, for ten days under 28 °C, and lastly, were scratched and incubated for 3–5 days under constant light to produce conidia. For hyphal growth, mycelia plugs were cultured on solid complete media (6 g of yeast extract, 6 g of casamino acid, 1 L of ddH_2_O, 20 g of agar) and kept for ten days in the incubator at 28 °C, and diameter measurements were taken, respectively. For protein, DNA, and RNA extraction, mycelia were oscillated in complete media for three days at 110 rpm under 28 °C. All the strains were kept on autoclaved dried papers for storage at 4 °C for further use.

### 2.2. Strategies for Generation of Mutant and Complementation Assays

To generate a MoPTC6 deletion mutant, a split-marker homologous recombination approach was deployed [26]. Plasmid vectors for knockout (split-marker) were constructed to delete *MoPTC6* in the *M. oryzae* genome. Flanking fragments A and B were amplified with appropriate primers designed, and both fragments, upstream and downstream, were separately ligated with hygromycin transferase (hph) on the pCX62 vector using overlapping polymerase chain reaction [27,28]. The PCR product used for the generation of deletion mutants was amplified for the constructed pCX62 vector using primer pair MoPTC6-BR + HY-F and MoPTC6-AF + YG-R. *M. oryzae* (Guy11) protoplast preparation and genetic transformation were conducted following the procedures described by [29,30,31]. Screening of putative transformants was performed by PCR using appropriate primers (MoPtc6_OR, MoPtc6_OF, and MoPtc6_UA) and a Southern blotting assay to confirm the positive candidates (mutant) [32]. For complementation assays, the MoPTC6 fragment comprising its native promoter and the open reading frame (ORF) genomic region was cloned behind GFP in the PKNTG vector. The *MoPTC6*-GFP construct was then transformed into ∆*Moptc6* protoplasts, transformants were selected by PCR, and GFP signals were visualized with a confocal microscope.

### 2.3. Southern Blotting Assay

Genomic DNA for Southern blot analysis was extracted following the cetyltrimethylammonium bromide (CTAB) method [33,34]. Genomic DNA quality was examined by running it on agarose, and the DNA concentration was measured using Nanodrop (Gene Group Company Ltd., Shanghai, China). The mutant and the WT genomic DNA were digested with the restriction enzyme (Cla I); this was followed by running the digested DNA on an agarose gel to separate the bands under 120 V for 10 min, which was then changed to 25 V overnight. The gel was denatured by soaking it in 0.5 M NaOH and transferred to a positively charged nylon membrane, which has binding capacity. The membrane was treated with a UV crosslinker, and prehybridization was performed under 42 °C for 45 min–2 h to block non-target sites on the membrane, which was followed by hybridization (probing the membrane) overnight to facilitate the binding of complementary bands in HL-2000 hybrilinker (LABrepco Ltd., Shanghai, China). Lastly, the membrane was washed with 2× SSC + SDS, maleic acid solutions. Further membrane washing was conducted following the starter kit I instructions and images were taken using the Tanon imaging system.

### 2.4. RNA Extraction and qPCR Assays

For RNA extraction, the WT Guy11 and ∆*Moptc6* mutant mycelia were cultured in complete media and oscillated at 110 rpm under 28 °C for 72 h. Mycelial samples were washed twice using double-distilled water and filtered using sterilized filter paper. Total RNA was extracted following the instructions of the Eastep super kit. cDNA was prepared by using an *Evo M-MLV RT* kit (Accurate Biotechnology, Changsha, China). The quantitative polymerase chain reaction (q-PCR) was performed using Cham Q universal SYBR qPCR master mix and Eppendorf Realplex2 master (Eppendorf AG 22331, Hamburg, Germany), and gene expressions were analyzed using the (2^−ΔΔCT^) formula [35].

### 2.5. Colony Growth, Conidiation Formation, and Conidiophore Development

To examine the fungal growth, mycelia plugs of the mutant Guy11 strain were spotted on solid complete media and kept in the incubator for 10 days at 28 °C. The diameters were then measured using a ruler. For conidiation assay, strains were spotted on conidia-inducing media (RBM), which is prepared by dissolving rice bran powder into 1 L of ddH_2_O and supplemented with agar powder to make it solid; the hyphae were then scratched off on day 10 after inoculation and incubated in a light chamber for an additional 72 h. Double-distilled water was used to wash conidia and count them under the microscope. To examine the conidiophore development, strains were grown on RBM (rice bran media); this was followed by scratching the medium and then cutting and transferring them to microscopic slides. Conidiophores were observed under the microscope at different time points (12 h, 24 h, and 36 h).

### 2.6. Evaluation of Pathogenicity, Hyphae Penetration, and Incipient Cytorryhsis Assays

For pathogenicity evaluation and rice infection, spores were harvested from fungi cultured on rice bran media by using double-distilled water containing 1% Tween 20 solution and then filtered with a double layer of clean paper. The concentration of conidia was adjusted up to 5 × 10^4^ conidia/mL and used to spray 3-week-old rice (*Oryzae sativa*). The sprayed plants were kept for 24 h in the dark, and then moved and incubated in the chamber where the light was on for 12 h and 12 h of dark cycle. Disease development was monitored every day, and images were taken after seven days of inoculation. For penetration and invasive development assays, the conidial suspension was applied to 10-day-old detached barley leaves, and then the sheath was peeled off. The invasive hyphae were examined, and photographs were taken using a Nikon microscope (Nikon corporation, Tokyo, Japan) at different time points, 30, 48, and 60 h, respectively. To study incipient cytorryhsis assay (cell collapse), 10 µL (5 × 10^4^) conidia/mL suspension was added to the surface that is hydrophobic, and kept in the incubator for 24 h. The water around the spores was sucked and then exchanged with an equal volume of 1 M, 2 M, and 3 M glycerol concentrations. The collapsed appressorium was recorded and viewed under a microscope for each replicate.

### 2.7. Cell Sensitivity Assay

To evaluate the influence of diverse stresses, such as osmotic cell walls, reagents were applied to ∆*Moptc6* and Guy11. Fungi were cultured on complete media containing 200 µg/mL Congo red, 0.01% SDS, 200 µg/mL CFW, 1 M NaCl, 1 M KCl, and 10 mM hydrogen peroxide (H_2_O_2_) separately to stress the fungus as well. Later, the mutant and WT were cultured and kept at 28 °C for ten days; subsequently, growth inhibition was calculated and photographs were taken.

### 2.8. Total Protein Extraction and Western Blotting Assays

To extract total protein, ∆*Moptc6* and Guy11 were inoculated in complete media and shaken at 110 rpm under 28 °C for 3 days. Then mycelia were harvested and ground under liquid nitrogen, and then collected into 2 mL tubes containing 1 mL of protein extraction buffer (containing 10 µL of proteinase inhibitor and 10 µL of phenylmethyl sulphonyl fluoride). Centrifugation was conducted at 14,000 rpm for 15 min at 4 °C. The upper clear supernatant liquid was pipetted carefully into 2 mL tubes, and SDS buffer was added. The proteins were then denatured at 100 °C for 10 min. The denatured proteins were analyzed by separating them on the SDS-PAGE and then transferred to a nitrocellulose paper membrane. The targeted proteins on the membrane were detected with P-p38 MAPK and P44/42MAPK primary antibodies, along with goat anti-rabbit and mouse IgG-HRP secondary antibodies. Actin mab was used as the reference antibody. A chemiluminescent detection reagent, Western brightTm (stabilized peroxide solution and lumino enhancer solution), was added and the signal was detected by a Western blotting kit (Advanta, München, Germany). Finally, photographs were taken using Tanon-5200 (Tanon Science& Technology Co., Ltd., Shanghai, China).

### 2.9. Microscopy Assays

MoPtc6 corresponding native promoters were fused in the C-terminus region of GFP and cloned in the pKNTG vector containing neomycin-resistant genes. The constructs were transformed into wild-type protoplasts, and the pictures were taken in growing hyphae, conidia, and appressoria. A Nikon confocal microscope (Nikon corporation, Tokyo, Japan) was used to visualize and image the green fluorescent protein (GFP). Conidia morphology, conidiophores development, appressorium development, and glycogen dynamics from conidium to appressorium were observed using Olympus DP80 (Olympus Co., Ltd., Beijing, China).

### 2.10. RNA Sequencing and Analysis

To extract RNA, mycelia derived from the ∆*Moptc6* mutant and the WT Guy11 strain were cultured in complete liquid media (CM) and shaken for 3 days at 110 rpm. The mycelia were then gathered by filtration using autoclaved filter paper and washed with double-distilled water. The wet mycelia were then dried by squeezing out water using dry autoclaved filters, and crushed by using a mortar and pestle in liquid nitrogen. The Eastep super kit was used for total RNA extraction. cDNA was prepared following the instructions of the *Evo M-MLV RT* kit. The cDNA library was sequenced and cleaned, and after that, reads were produced. Lastly, the alignment of reads to the reference genome of *M. oryzae* by using Hist2 was performed [36]. Differentially expressed genes (DEGs) were examined by DESeq. Further analysis was carried out by Gene Ontology and KEGG [37].

### 2.11. Statistical Analysis

GraphPad Prism 8 was used for statistical analysis. The data from biological replicates were analyzed using one-way ANOVA (including nonparametric tests). ImageJ software (Version 1.54P) was utilized for densitometric analysis of the Western blot bands.

## 3. Results

### 3.1. Identification, Domain Architecture, and Phylogenetic Analysis of MoPtc6

To identify MoPtc6 in *Magnaporthe oryzae*, we performed a homology BLAST search using the Ptc6 protein sequence from *Saccharomyces cerevisiae* (https://blast.ncbi.nlm.nih.gov/Blast.cgi, accessed on 20 February 2025). The BLASTp hit reveals the *MoPTC6* (MGG_03918) gene in the *M. oryzae* genome. Domain prediction and architecture revealed that the PP2C_SIG domain is present in six MoPtc6 orthologs, while the PP2Cc domain is conserved in five orthologs (Figure 1a). Phylogenetic analyses indicated that MoPtc6 shares the same ancestry as its orthologs in different fungi, including *Fusarium gramineum*, *Fusarium oxysporum*, *Neurospora crassa,* and *Botrytis cinerea,* displayed an identity of 83% with *N*. *crassa* and 97% with *F. graminearum*, *F. oxysporum,* and *B. cinerea* (Figure 1b). Next, we generated the 3D structure of MoPtc6 from the Swiss Model Software (https://www.genecards.org/, accessed on 20 February 2025) (Figure 1c). Overall, these results suggest that the Ptc6 protein phosphatase is conserved in pathogenic fungi.

### 3.2. Expression Profile of MoPTC6 During Pathogen–Host Interaction and Subcellular Localization

To explore the importance of the *MoPTC6* gene on fungal development. We first checked its expression during host–pathogen interaction. Conidia were harvested from WT Guy11 and used to spray on rice seedlings. Samples were collected at different time points (4 h, 8 h, 12 h, 24 h, 48 h, and 72 h), and total RNA was extracted for qRT-PCR assay. The expression of *MoPTC6* was upregulated at 48 and 72 h post-inoculation (hpi), suggesting that *MoPTC6* could be crucial in hyphae invasion of host tissues (Figure 2a). We next investigated the co-localization of MoPtc6-GFP by using a confocal scanning microscope. Results showed MoPtc6 is sub-localized in the cytoplasm of conidia, mycelia, and in appressoria (Figure 2b).

### 3.3. Effects of MoPTC6 Deletion on Fungal Growth, Conidiophore Development, and Sporulation

To investigate the roles of MoPtc6 in fungal growth and conidiation of blast fungus, we monitored the growth of ∆*Moptc6* and WT Guy11 on CM medium. Growth of ∆*Moptc6* was remarkably attenuated compared to the WT strain (Figure 3a,b). For conidiation, we harvested and quantified the amount of conidia generated by the mutant and WT grown on rice bran media and found that the ∆*Moptc6* mutant generated a few conidia (Figure 3c). Also, conidiophore assay revealed that the ∆*Moptc6* produces fewer conidiophore structures than the WT Guy11 strain; the mutant conidiophores bear one or two conidia, unlike those of the WT Guy11 that had a bunch of conidia per conidiophore (Figure 3d). Moreover, we analyzed the expression levels of conidiation-encoding genes such as *MoCON1*, *MoCON6*, *MoCON7*, *MoFLBA*, and *MoFLB,* and the results indicated that all the genes were downregulated (Figure 3e). Overall, these findings revealed that MoPtc6 is important for conidiation and growth rate in *M. oryzae*.

### 3.4. Sensitivity of ΔMoptc6 Mutant to Stress-Inducing Agents

To evaluate the role of MoPtc6 on stress resistance, we cultured the WT, Δ*Moptc6* mutant, and Δ*Moptc6*_C strains on CM media with 200 µg/mL Congo red (CR), 200 µg/mL Calcofluor white (CFW), 1 M sodium chloride (NaCl), 10 mM hydrogen peroxide (H_2_O_2_), 1 M potassium chloride (KCl), and 0.02% (SDS) as supplements. The Δ*Moptc6* showed significant growth inhibition compared to the WT Guy11 due to the stress agents (Figure 4a,b). These findings imply that MoPtc6 is important for cell wall integrity and osmoregulation. To further unveil the mechanism of stress resistance by MoPtc6, we examined the phosphorylation levels of stress-regulating proteins Mps1 and Osm1 in the various strains. Western blotting assay revealed that Osm1 phosphorylation decreased in ∆*Moptc6* (Figure 4c,d). However, the phosphorylation level of Mps1 was increased in the ∆*Moptc6* mutant in comparison to the WT (Figure 4e,f), suggesting the crucial role of MoPtc6 in the phosphorylation level of Mps1 and Osm1 proteins in blast fungus.

### 3.5. Deletion of MoPTC6 Attenuates Appressorium Turgor Pressure

For successful penetration, the blast fungus develops a critical structure called an appressorium on the plant surface, which accumulates high turgor pressure used to forcefully puncture and penetrate the harder cuticle layer of the host [38]. To investigate the cause of reduced virulence in ∆*Moptc6*, we investigated and compared the appressorium turgor pressure of the various strains by harvesting fresh spores from the fungal strains and subjecting them to an incipient cytorryhsis assay. Most of the Δ*Moptc6* mutants were found to produce collapsed appressoria with significantly reduced turgor pressure as indicated by their percentage of incipient cytorryhsis (Figure 5a,b). Because turgor generation depends on glycogen mobilization from conidia to appressoria and its subsequent degradation in the appressoria [39], we monitored the spatiotemporal mobilization and degradation of glycogen at different time points by potassium iodide (KI) staining. Within 24 h of KI staining, all the fungal strains had glycogen accumulated in their appressoria (Figure 5c,d). From 8 h to 12 h, and up to 24 h of staining, the glycogen contents of these appressoria reduced with time, but the Δ*Moptc6* mutant significantly retained more glycogen in the appressoria than the WT and Δ*Moptc6*_C strains, indicating a weaker glycogen degradation rate in the mutant, which results in weak turgor generation.

### 3.6. ∆Moptc6 Plays a Significant Role in M. oryzae Virulence

To analyze the influence of MoPtc6 in the pathogenicity of *M. oryzae*, we sprayed rice seedlings with conidia suspension derived from the Δ*Moptc6* mutant, Guy11, and the complemented strain. At 1-week post-infection, the Δ*Moptc6* leaves displayed a significantly reduced pathogenicity (Figure 6a,b). Similarly, we infected barley with mycelial plugs of Δ*Moptc6* and WT strains, and the results revealed that the infection rate of the ∆*Moptc6* was decreased (Figure 6c,d). We next evaluate the fungal hyphal invasion within barley epidermal cells at different time points, 24 h, 48 h, and 72 h post-inoculation. Results demonstrated that loss of *MoPTC6* weakened the fungal invasive growth in the barley cells (Figure 6e). Taken together, this result indicates that *MoPTC6* is required in the pathogenesis of *M. oryzae*.

### 3.7. Influence of MoPTC6 Gene Deletion on Global Gene Expression

Due to the significant phenotype resulting from the loss of *MoPTC6*, we conducted RNA sequencing of the *∆Moptc6* compared to the WT (Guy11). This assay aimed to gain insight into differentially expressed genes (DEGs), GO, and KEGG that are enriched by expressed genes upon deletion of *MoPTC6*. The gene expression analysis revealed that 8533 genes were expressed in both the WT and *∆Moptc6* mutant, with 825 and 489 genes expressed in the WT and ∆*Moptc6* (Figure 7a,b). Next, we analyzed all DEG-based GO enrichments (*p* ≤ 0.01). A total of 43 GO terms were enriched in biological processes (17 terms), cellular components (14 terms), and molecular functions (12 terms) (Appendix A). The KEGG enrichments of DEGs were also analyzed, and the results showed that many pathways were associated with metabolic processes (Figure 7c).

## 4. Discussion

Type 2C protein phosphatases are important for diverse signaling and biological pathways in eukaryotes. Furthermore, their contributions to the physiology and development of *M. oryzae* remain poorly analyzed. The various PP2Cs have already been discussed in *Saccharomyces cerevisiae* [40,41]. The number of phosphatases identified in *M. oryzae* is consistent with other filamentous model fungi, including *Fusarium oxysporum*, *Botrytis cinerea*, and *Fusarium gramineurum* [14,15,40]. In this work, we explored the functions of PP2Cs MoPtc6 in fungal growth, stress tolerance, and virulence of rice blasts. Firstly, we evaluated the transcription expression pattern of MoPtc6 in different developmental stages of the disease cycle and found that MoPtc6 was up-regulated at 48 and 72 h post-inoculation (hpi). Indicating a possible role in host colonization during plant–pathogen interaction. Since the localization of a protein is related to function, we decided to check the localization of MoPtc6 at all fungal developmental stages, and the findings demonstrated that MoPtc6 is sub-localized in the cytoplasm of hyphae, spores, and appressoria. These findings suggest that MoPtc6 might be required in different subcellular processes during fungal development and pathogenesis.

Conidiation is crucial for the pathogenesis of filamentous fungi such as *Fusarium averaceum*, *Neurospora crassa*, and, *Aspergillus nidulans* [42,43]. Previous studies demonstrated that protein phosphatases are involved in asexual reproduction in *M. oryzae* [44]. This study further strengthens the involvement of protein phosphatases in the reproduction process of rice blast fungus, as the ∆*Moptc6* produced few spores and conidiophore structures compared to the WT Guy11 strain, which might be due to the weak expression of conidiation-encoding genes [45].

Abiotic stresses, such as osmotic and oxidative stressors, were previously discussed and elaborated in *S. cerevisiae* [46,47]. Cell wall maintenance pathways are required in the stress adaptation mechanisms, cell anatomy, and virulence of *M. oryzae* [48]. In this work, the ∆*Moptc6* growth was strongly inhibited by cell wall stress agents such as Congo red compared to the WT; thus, indicating that *MoPTC6* is important for tolerance to the cell wall stress reagents in *M. oryzae* development, which is in agreement with previous studies cited [49,50]. Ultimately, our findings revealed that the mutant exhibited defects in osmotic stresses, including NACL and KCL; this suggests the involvement of Ptc6 in responses to hyperosmotic stress in *M. oryzae* and is consistent with [51,52,53]. The variation of antioxidants and free radicals leads to oxidative stress in eukaryotes [54]. Furthermore, disruption of *MoPTC6* increased sensitivity towards oxidative stress, such as hydrogen peroxide (H_2_O_2_); this means that Ptc6 could probably be recruited in the regulation of fungal oxidant-sensing pathways [55,56]. Osm1, which is commonly known as Hog1 protein, is required in stress responses and phosphorylated under diverse osmotic, oxidative stresses, and fungicides [57,58,59]. Various cell wall stresses regulate Mps1 phosphorylation and the MAP kinase pathway, as discussed earlier [60,61,62,63]. Moreover, Western blot assay demonstrated that deletion of MoPtc6 is important in the protein post-modification and phosphorylation of the MoMps1 and MoOsm1 in *M. oryzae*; these findings are in line with [15,64]. We suggest that *MoPTC6* enhances the activation of Slt2-mediated cell wall maintenance and regulates proteins responsible for stress in *M. oryzae*. Taken together, *MoPTC6* plays a key function in responses to the abiotic and biotic stresses, and homeostasis in *M. oryzae* development.

Appressoria turgor pressure is induced by a higher concentration of metabolites, which is essential for the penetration and invasion of fungal hyphal into the host tissue [65,66,67]. Here, we demonstrated through biochemical assay (incipient cytorryhsis) that *MoPTC6* potentially is important for appressorium turgor pressure, which was confirmed by many appressoria collapsing in ∆*Moptc6* after being treated with a diverse concentration of glycerol. Moreover, ∆*Moptc6* delayed glycogen transport and degradation from conidium to appressorium, which showed that MoPtc6 is exclusively required in the trafficking of glycogen utilization in *M. oryzae*. Our results are in line with previous publications that detailed appressorium turgor during the infection process in *M. oryzae* [68,69]. Here, we presumed that *MoPTC6* participated in the glycogen metabolism and turgor pressure maintenance in *M. oryzae*.

For host infection, many filamentous fungi, including rice *M. oryzae*, need to develop a structure called an appressorium to gain entry into host organisms [33]. Through a comprehensive analysis of ∆*Moptc6*, findings led to the reduction of virulence in rice and barley, and these findings are in line with an early report on *M. oryzae* [70]. From these results, we proposed that *MoPTC6* may serve as an essential factor for pathogenicity and related pathways, including MAP kinase pathways, as regulators of fungal virulence. MAP kinases are dual phosphorylated protein kinases, present in eukaryotes. Taken together, we suggested that PP2Cs are involved in the virulence mechanisms of *M. oryzae*.

## 5. Conclusions

Overall, we employed the filamentous fungus *M. oryzae* as a model pathosystem to identify and characterize the protein phosphatase MoPtc6 and revealed that the activities of MoPtc6 contribute to fungal virulence, conidia production, vegetative growth, appressorium turgor pressure generation, multi-stress tolerance, phosphorylation of MoOsm1, MoMps1, and transcript expression level. Based on the findings, we expect that our findings will provide new insights to combat blast fungus. Further work entails to identification and characterization of substrate proteins by using molecular and biochemical tools and revealing their functional roles in the fungal development and pathogenesis of *M. oryzae*.

## Figures and Tables

**Figure 1 jof-11-00335-f001:**
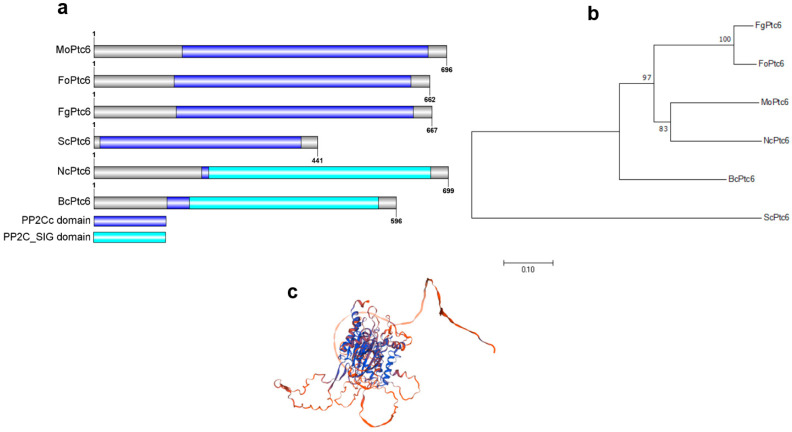
Domain structure and phylogenetics of MoPtc6 and its orthologs. (**a**) The domain architecture of Ptc6 proteins from different fungi (PP2Cs domain and PP2C_SIG domain). The architecture was built using IBS software (IBS.biocuckoo.org, accessed on 20 November 2024). (**b**) Analysis of the phylogenetic relationship of Ptc6 proteins among different pathogenic fungi. The tree was produced by using MEGA X software version 10.1. (**c**) Three-dimensional structure of MoPtc6 generated from Swiss Model Software https://www.genecards.org/, accessed on 20 November 2024. Mo: *Magnaporthe oryzae*; Fo: *Fusarium oxysporum*; Fg: *Fusarium gramineurum*; Sc: *Saccharomyces Cerevisiae*; Bc: *Botrytis Cinerea*. Nc: *Neurospora crassa*.

**Figure 2 jof-11-00335-f002:**
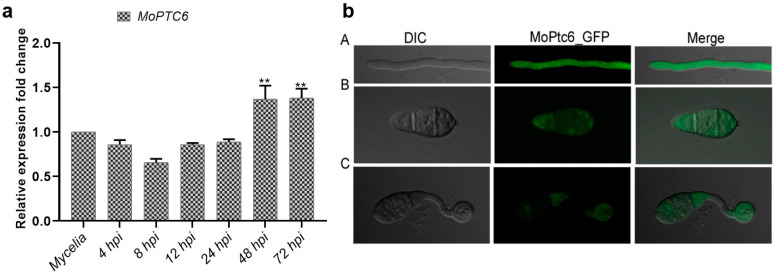
Expression pattern and localization of *MoPTC6* in *M. oryzae*. (**a**) Transcription patterns of *MoPTC6* during interaction with the host; three-week-old rice seedlings were sprayed with Guy11, and the samples were harvested at different time points (4 h, 8 h, 12 h, 24 h, 48 h, and 72 h). (**b**) Sub-localization of MoPtc6-GFP in the mycelia, conidia, and appressoria. Images were taken by using confocal scanning microscope. (** *p* < 0.01).

**Figure 3 jof-11-00335-f003:**
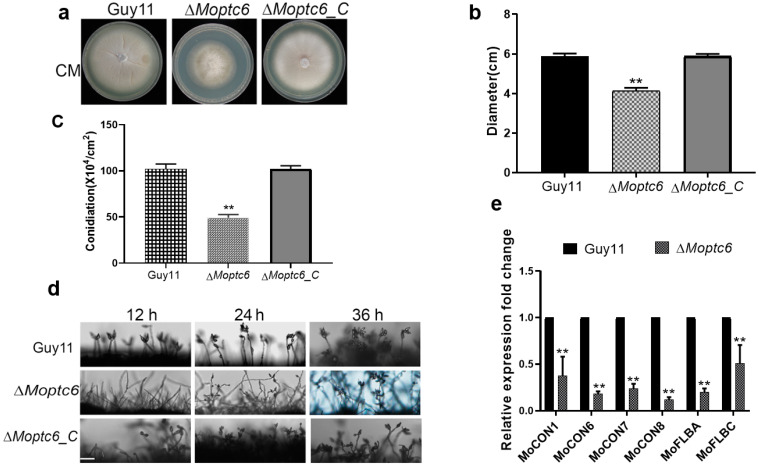
MoPtc6 is important for growth and conidiation in *M. oryzae*. (**a**,**b**) Growth and colony diameters of the mutant and WT grown on solid complete media at 28 °C for ten days. (**c**,**d**) Quantification of conidia and conidiophores development in the ∆*Moptc6* and WT. (**e**) Expression levels of conidiation-related genes in ∆*Moptc6* compared to the WT. Asterisks show statistically significant differences analyzed by ANOVA (**, *p* < 0.01). Scale bar = 20 μm.

**Figure 4 jof-11-00335-f004:**
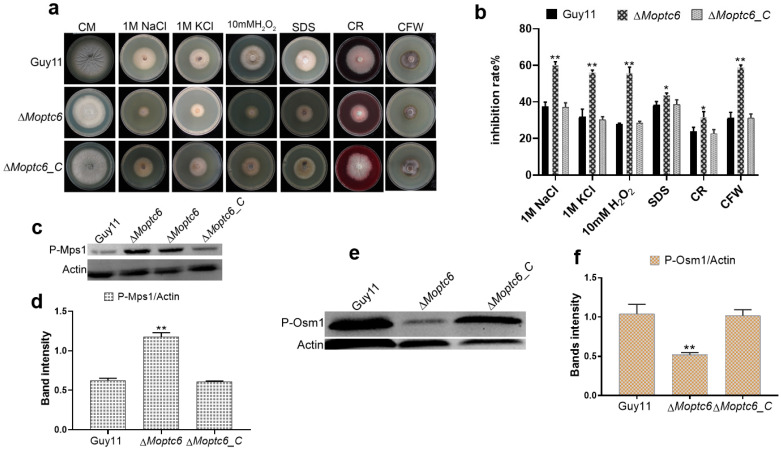
Sensitivity of ∆*Moptc6* mutant and WT to various stresses. (**a**) Growth of the strains on CM media with 200 µg/mL CR, 0.01% SDS, 200 µg/mL CFW, 1 M KCl, 1 M NaCl, and 10 mM H_2_O_2_ at 28 °C for ten days. (**b**) Inhibition of growth of the strains due to the diverse fungal stressors. (**c**,**d**) Osm1 phosphorylation levels; MoOsm1 was bound to P38 Thr180/Tyr182 antibody and actin was used as a control. Quantification of band intensity was performed with ImageJ software. (**e**,**f**) Mps1 phosphorylation levels; MoMps1 were detected with antiphospho P44/42 antibody, and actin was used as a control. (**f**) Displayed bands’ intensity in mutant and WT. Asterisks show statistically significant differences (*, *p* < 0.05; **, *p* < 0.01).

**Figure 5 jof-11-00335-f005:**
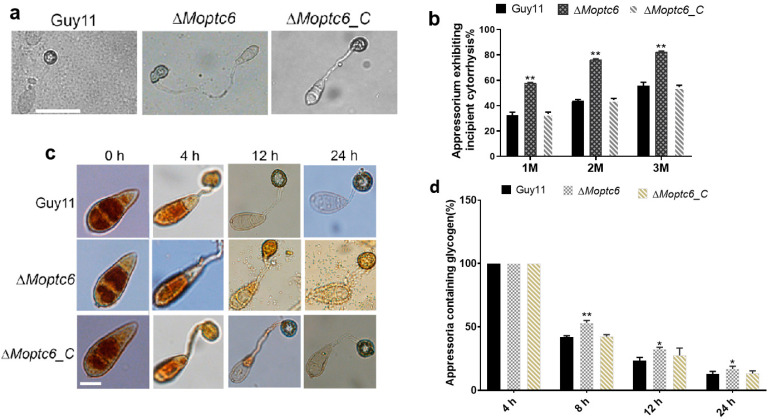
∆*Moptc6* promotes appressorium turgor pressure and mobilization of glycogen. (**a**,**b**) Indicates incipient cytorryhsis conducted to examine appressorium turgor pressure for ∆*Moptc6*, WT, and complementation strains. Appressoria were incubated for 24 h period and tested with various concentrations of glycerol, 1, 2, and 3 M, 5 min before viewed by using microscope. Scale bar = 20 μm. (**c**,**d**) Micrograph displayed glycogen utilization from conidium to appressorium and percentages of glycogen levels in conidium and in appressorium at different time points, 4 h, 8 h, 12 h, and 24 h, in mutant compared to WT. Asterisks displays statistically significant (*, *p* < 0.05; **, *p* < 0.01). Scale bar = 10 μm.

**Figure 6 jof-11-00335-f006:**
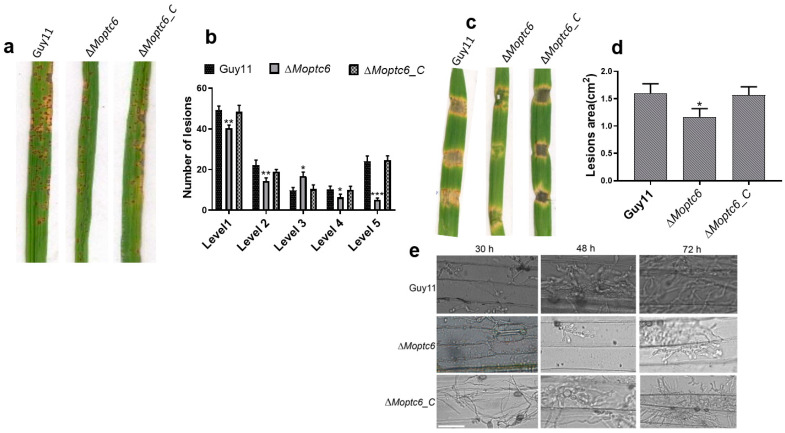
Virulence analysis of MoPtc6 in *M. oryzae*. (**a**,**b**) The figures displayed the pathogenicity analysis and lesions quantification assay for ∆*Moptc6*, WT, and complementation strain. Images were captured one-week post-spraying. (**c**,**d**) Indicate the lesion areas on barley leaves induced by Δ*Moptc6*, Δ*Moptc6-com*, and the WT strains’ mycelia plugs; lesion areas were calculated by using Image J software. (**e**) Observation of invasive hyphae in ten-day-old barley leaves infected with 5 × 10^4^ spores/mL and photographed after 24 h, 48 h, and 72 h post-inoculation. Asterisks displays statistically significant (*, *p* < 0.05; **, *p* < 0.01; ***, *p* < 0.001). Scale bar = 20 μm.

**Figure 7 jof-11-00335-f007:**
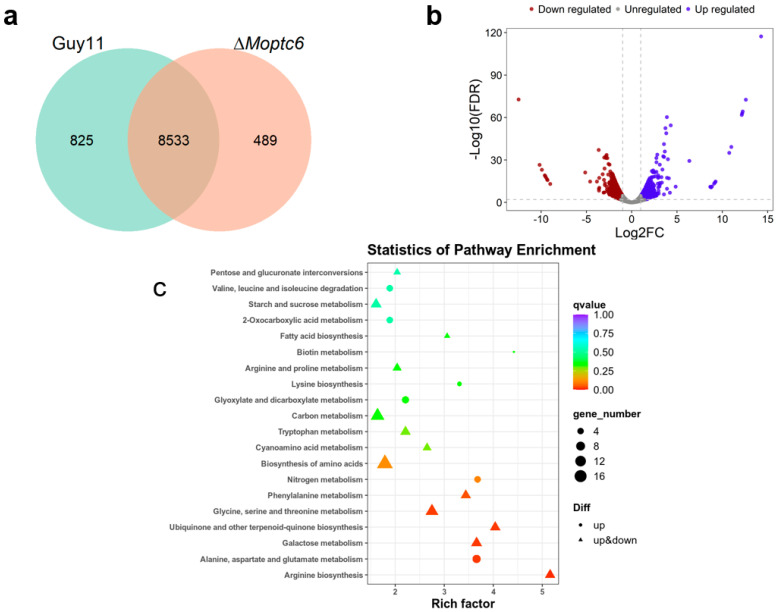
Expression of genes between ∆*Moptc6* and WT. (**a**) Global view of gene expression levels in ∆*Moptc6* and WT. This figure was generated by Jvenn software version 2.1.0. (**b**) Volcano figure plotted by using R software 4.3.3, indicating differentially expressed gene (DEG) distribution patterns in ∆*Moptc6* mutant and WT. Red dots demonstrate downregulated genes, and blue dots represent up-regulated genes. (**c**) Displaying the KEGG pathways enriched by differentially expressed genes (DEGs) (*p* < 0.01).

## Data Availability

The original contributions presented in this study are included in the article/Appendix A. Further inquiries can be directed to the corresponding authors.

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
