# Peer review of "Type 2C Protein Phosphatase MoPtc6 Plays Critical Roles in the Development and Virulence of Magnaporthe oryzae"

_jof, 2025, doi:10.3390/jof11050335_

Round 1
Reviewer 1 Report
The article is devoted to the study of the pathogenicity mechanisms of Magnaporthe oryzae, one of the most dangerous phytopathogens threatening global rice production. Uncovering the role of protein phosphatase MoPtc6 in the development and virulence of the fungus is important for the development of plant protection strategies. The authors used a fairly comprehensive approach, including methods of functional genomics, biochemistry and microscopy, which allows for a comprehensive characterization of the role of MoPtc6. Particularly valuable are the data on phosphorylation of key proteins (MoOsm1, MoMps1) and transcriptome analysis.
The article presents fairly clear results, such as, for example, the deletion of the MoPTC6 gene leads to significant disturbances: decreased hyphal growth, conidia formation, stress resistance, weakening of the turgor pressure of appressoria and virulence. This confirms the key role of the protein in pathogenesis.
The practical significance is monitored, since the results can be used to search for new fungicide targets or to create resistant rice varieties.
However, some limitations and questions remain in my opinion.
Although the effect of MoPtc6 on the phosphorylation of MoOsm1 and MoMps1 is shown, the details of these interactions remain unclear. Further studies, such as the identification of MoPtc6 substrates, are needed.
In my opinion, the depth of transcriptome analysis is limited. The enrichment of KEGG pathways is schematically presented; a detailed analysis of key virulence-related genes is missing.
There is a possible lack of in vivo data. Plant experiments are limited to symptom assessment; analysis of early stages of infection (e.g., host immune response) could enhance the work.
In summary, this paper represents a significant contribution to the understanding of the molecular basis of M. oryzae pathogenicity. The data obtained convincingly demonstrate the role of MoPtc6 in fungal development and its interaction with the host plant. However, further studies, including the identification of its substrates and signaling pathways, are required to fully elucidate the mechanisms of action of MoPtc6. The work has high scientific value and can become the basis for applied solutions in the fight against rice blast.
The article is devoted to the study of the pathogenicity mechanisms of Magnapor the oryzae, one of the most dangerous phytopathogens threatening global rice production. Uncovering the role of protein phosphatase MoPtc6 in the development and virulence of the fungus is important for the development of plant protection strategies. The authors used a fairly comprehensive approach, including methods of functional genomics, biochemistry and microscopy, which allows for a comprehensive characterization of the role of MoPtc6. Particularly valuable are the data on phosphorylation of key proteins (MoOsm1, MoMps1) and transcriptome analysis.
The article presents fairly clear results, such as, for example, the deletion of the MoPTC6 gene leads to significant disturbances: decreased hyphal growth, conidia formation, stress resistance, weakening of the turgor pressure of appressoria and virulence. This confirms the key role of the protein in pathogenesis.
The practical significance is monitored, since the results can be used to search for new fungicide targets or to create resistant rice varieties.
However, some limitations and questions remain in my opinion.
Although the effect of MoPtc6 on the phosphorylation of MoOsm1 and MoMps1 is shown, the details of these interactions remain unclear. Further studies, such as the identification of MoPtc6 substrates, are needed.
In my opinion, the depth of transcriptome analysis is limited. The enrichment of KEGG pathways is schematically presented; a detailed analysis of key virulence-related genes is missing.
There is a possible lack of in vivo data. Plant experiments are limited to symptom assessment; analysis of early stages of infection (e.g., host immune response) could enhance the work.
In summary, this paper represents a significant contribution to the understanding of the molecular basis of M. oryzae pathogenicity. The data obtained convincingly demonstrate the role of MoPtc6 in fungal development and its interaction with the host plant. However, further studies, including the identification of its substrates and signaling pathways, are required to fully elucidate the mechanisms of action of MoPtc6. The work has high scientific value and can become the basis for applied solutions in the fight against rice blast.
Author Response
The article is devoted to the study of the pathogenicity mechanisms of Magnaporthe oryzae, one of the most dangerous phytopathogens threatening global rice production. Uncovering the role of protein phosphatase MoPtc6 in the development and virulence of the fungus is important for the development of plant protection strategies. The authors used a fairly comprehensive approach, including methods of functional genomics, biochemistry and microscopy, which allows for a comprehensive characterization of the role of MoPtc6. Particularly valuable are the data on phosphorylation of key proteins (MoOsm1, MoMps1) and transcriptome analysis. The article presents fairly clear results, such as, for example, the deletion of the MoPTC6 gene leads to significant disturbances: decreased hyphal growth, conidia formation, stress resistance, weakening of the turgor pressure of appressoria and virulence. This confirms the key role of the protein in pathogenesis.
The article presents fairly clear results, such as, for example, the deletion of the MoPTC6 gene leads to significant disturbances: decreased hyphal growth, conidia formation, stress resistance, weakening of the turgor pressure of appressoria and virulence. This confirms the key role of the protein in pathogenesis.
The practical significance is monitored, since the results can be used to search for new fungicide targets or to create resistant rice varieties.
However, some limitations and questions remain in my opinion.
Although the effect of MoPtc6 on the phosphorylation of MoOsm1 and MoMps1 is shown, the details of these interactions remain unclear. Further studies, such as the identification of MoPtc6 substrates, are needed.
Answer: Thank you for this suggestion, type 2C protein phosphatases is a larger family, we have a plan to find all substrates in our next studies, we hope that our request will be taken into your consideration.
In my opinion, the depth of transcriptome analysis is limited. The enrichment of KEGG pathways is schematically presented; a detailed analysis of key virulence-related genes is missing.
Answer: Thank you for this suggestion, in this current we wanted to make aware the readers only about Global view of genes expression levels in ∆Moptc6 and WT and KEGG pathways enriched by differential expressed genes
There is a possible lack of in vivo data. Plant experiments are limited to symptom assessment; analysis of early stages of infection (e.g., host immune response) could enhance the work.
Answer: Thank you for this suggestion, in vivo we showed how fungal invasive hyphae penetrated and invaded the plant cuticle, we promise that in our next studies on Protein phosphatase we will do experiments on host immune responses during infection with Magnaporthe oryzae.
Reviewer 2 Report
The manuscript describes type 2C protein phosphatase MoPtc6 in the fungal pathogen Magnaporthe oryzae. Modern molecular techniques were used in the study (qRT-PCR, sequencing, Southern Blotting Assay) to analyze RNA sequences (gene expression) and determine protein content. Pathogenicity tests of wild-type and mutant isolates were also performed, using microscopy techniques to analyze the infection process.
The specific comments listed below should be addressed to make the results of the study more understandable.
Numerous unnecessary spaces should be eliminated throughout the manuscript.
- 32-32: Please describe the infection process of M. oryzae and spore formation by this pathogen.
- 34: Please list the countries where this disease is most devastating and causes the greatest losses.
- 75: It should be “H2O” – please correct.
- 76: Where the plates with solid media shaken? For what purpose?
L.125: The composition of RBM medium should be provided.
Subsection 2.6: How was the infected leaf area measured? The results are expressed in cm2.
- 211: The species name Fusarium oxysporum should be italicized.
Figure 1. The “Nc” abbreviation should be defined. It appears that Figure 1A illustrates the structure of Neurospora crassa protein.
Figure 2B: At which stage of infection (how many days after inoculation) were the microscopic images taken?
The analyzed phenomena are well explained in the Discussion section, but they should be presented in a broader context, including e.g. the description of the deleterious effects of M. oryzae infection.
Author Response
The specific comments listed below should be addressed to make the results of the study more understandable. Numerous unnecessary spaces should be eliminated throughout the manuscript.
- Line 32-32: Please describe the infection process of M. oryzae and spore formation by this pathogen
Answer: Thank you for this comment, Conidia are formed from conidiogenous cells by a blastic mode of development, infection begins when a three-celled conidium lands and attaches to the hydrophobic surface of a rice leaf. The spore germinates producing a long narrow germ tube that differentiates into an appressorium which subsequently penetrate and invade the plant cuticle.
2 . Line 34: Please list the countries where this disease is most devastating and causes the greatest losses.
Answer: Thank you for this suggestion, Rice blasts are reported to cause considerable harvest losses, particularly in Asia, namely China, Sri Lanka, Indonesia, Bangladesh and India. In addition, the disease is also a major problem in Brazil, Bolivia, Australia, Korea and the Philippines.
3.75: It should be “H2O” – please correct.
Answer: Thank you for this comment, change was made as well
4.76: Where the plates with solid media shaken? For what purpose?
Answer: Thank you for this important comment, we made correction of the whole section 2.1 of the materials and Methods
L.125: The composition of RBM medium should be provided.
Answer: Thank you for this observation, Rice bran media (RBM) was prepared by dissolving rice bran powder into1L of ddH2O and supplemented with agar powder to make it solid
Subsection 2.6: How was the infected leaf area measured? The results are expressed in cm2.
Answer: Thank you for this observation, here we used software called ImageJ to calculated the infection Area in cm2, this software has plotting scale to measure the area.
211: The species name Fusarium oxysporum should be italicized.
Answer: Thank you we made correction accordingly
Figure 1. The “Nc” abbreviation should be defined. It appears that Figure 1A illustrates the structure of Neurospora crassa protein.
Answer: Thank you we made correction as suggested
Figure 2B: At which stage of infection (how many days after inoculation) were the microscopic images taken?
Answer: Thank you for question, we added more description in the materials and methods section 2.9. we checked the protein sub localization in fungus not in the plantae. MoPtc6 corresponding native promoters, were fused in the C‐terminus region of GFP and cloned in the pKNTG vector containing neomycin‐resistant genes constructs were transformed into wild type protoplast. Results obtained showed that MoPtc6‐GFP targeted to the cytoplasm in growing hyphae, conidia, and appressorium. In hyphae before we took pictures the transformants were picked and grown on CM media for 4-7 days. We next cultured the fungi on Rice bran media to produce conidia and the picture was taken, Lastly, we dropped conidia solution on hydrophobic slides and incubated for 8-24 hours under 28oC to generate appressium.
The analyzed phenomena are well explained in the Discussion section, but they should be presented in a broader context, including e.g. the description of the deleterious effects of M. oryzae infection.
Answer: Thank you for this comment, we have added some information to improve the statement line 404.